# Role of MicroRNA-7 (MiR-7) in Cancer Physiopathology

**DOI:** 10.3390/ijms23169091

**Published:** 2022-08-13

**Authors:** Mario Morales-Martínez, Mario I. Vega

**Affiliations:** 1Molecular Signal Pathway in Cancer Laboratory, UIMEO, Oncology Hospital, Siglo XXI National Medical Center, IMSS, Mexico City 06720, Mexico; 2Department of Medicine, Hematology-Oncology Division, Greater Los Angeles VA Healthcare Center, UCLA Medical Center, Jonsson Comprehensive Cancer Center, Los Angeles, CA 90095, USA

**Keywords:** miR-7, cancer, cancer progression, clinical implication, gene expression

## Abstract

miRNAs are non-coding RNA sequences of approximately 22 nucleotides that interact with genes by inhibiting their translation through binding to their 3′ or 5′ UTR regions. Following their discovery, the role they play in the development of various pathologies, particularly cancer, has been studied. In this context, miR-7 is described as an important factor in the development of cancer because of its role as a tumor suppressor, regulating a large number of genes involved in the development and progression of cancer. Recent data support the function of miR-7 as a prognostic biomarker in cancer, and miR-7 has been proposed as a strategy in cancer therapy. In this work, the role of miR-7 in various types of cancer is reviewed, illustrating its regulation, direct targets, and effects, as well as its possible relationship to the clinical outcome of cancer patients.

## 1. Introduction

Cancer is defined by the WHO as a broad group of diseases that can affect any part of the body, characterized by the rapid proliferation of abnormal cells that extend beyond their usual limits and can invade adjacent parts of the body or spread to other organs through a process called metastasis. Metastases are the leading cause of cancer death [1].

Cancer is influenced by several factors including the activation of proto-oncogenes through chromosomal translocations such as the presence of the Philadelphia chromosome, which is the result of the translocation of the BCR-ABL genes present in most patients with chronic myeloid leukemia (CML) [2]. Additionally, mutations can inactivate genes that give rise to cancer suppressors [3], for example, the mutation and consequent inactivation of p53, which leads to an accumulation of mutant protein, making it a hallmark of cancer [4]. Several miRNAs have been functionally classified as oncomiRs or tumor repressors because they act as transcriptional regulatory factors, depending on the target they regulate [5].

### 1.1. Emergence of MiRNAs

In 1993, Dr. Ambros’ group described for the first time the presence of a small non-coding RNA in the lin-4 locus that induced inhibition of lin-14 protein expression; lin-14 is a protein involved in controlling the transition between the initial larval stages of the nematode life cycles and is characterized by a complementary sequence to the 3′ UTR region of lin-14 [6]. These findings were confirmed by Dr. Gary Ruvkun’s group, who described the pairing between the non-coding transcript of lin-4 and the 3′ UTR sites of lin-14 [7]. These groups laid the foundations for the current study of microRNAs (miRNAs), a term that was coined in 2001 by Dr. Víctor Ambros [8], who, through the use of bioinformatics, also reported that these molecules could be numerous and diverse [9]. Later studies demonstrated that many of these miRNAs exist in invertebrates and vertebrates, and some RNAs that are similar to miRNA let-7, characterized by *Caenorhabditis elegans*, are highly conserved. These findings suggested a new mechanism of post-transcriptional regulation mediated by small RNAs [10] that could participate in the regulation of many signaling pathways [8], and they are expressed much more frequently than previously reported. The miRNAs are expressed in different species, such as mice [11], humans [12,13], and flies [14], complementing what was previously reported [15], in *C. elegans* [9]

### 1.2. MiRNAs Biogenesis

One mature microRNA can be codified in different sites in the genome; for example, lin-4 and let-7 are derived from independent transcription units [10]. On the other hand, some miRNAs are found in pre-mRNA, although these are not the most common [15]. Interestingly, some miRNAs are found in groups, and their pattern implies a multicistronic-type transcription [10]. In particular, most microRNA genes in humans are isolated [16].

miRNAs are synthesized in the cell nucleus from a long primary miRNA transcript (pri-miRNA) and subsequently processed by the ribonucleases Drosha and DGCR8/Pasha, a type III RNase [17], generating a precursor transcript or pre-miRNA [18]. The structure that results from the processing by Drosha has as its characteristic an end that stands out in the 5′ region from ~2 to 3 nt [19]. The pre-miRNA with the 5′ overhang end is recognized by the Exportin 5 complex, which exports the pre-miRNA to the cytosol, where it is processed by the ribonuclease Dicer in a complex formed by the TRBP and PACT proteins [20], generating miRNA doublets approximately 22 nucleotides long [21]. Subsequently, a chain of the doublet that corresponds to the presence of the 5′ overhang [22,23] is deposited within a protein called Argonaute (Ago), generating the ribonucleoprotein inhibitor complex (miRISC) originating a functional mature miRNA (Figure 1). Previously, it was thought that the other chain of the doublet was degraded; however, different reports have provided evidence that it has the functional activity of regulating genes [24]. The assembly of the miRISC is associated with specific messenger RNAs based on their sequence complementarity (consensus region) with the 3′ non-coding region of the mRNA (3′-UTR), coding sequences, or the 5′ non-coding region (5′-UTR). The miRISC can be identified based on the perfect or semi-perfect complementary pairing between the microRNA and the mRNA [25], and this association regulates the expression of the gene at the post-transcriptional level by suppressing either the stability, the translation of the mRNA, or both, depending on its complementarity with the binding site or target site. If the complementarity is sufficient, a cut is induced; however, if the complementarity is partial, the cut is not induced, and it will be located in the complementary sites and translation will be prevented by inhibition in the ribosome [26,27,28]. The mechanism is illustrated in Figure 1.

Currently, 2300 true miRNAs have been described in the human genome [29], some of which are similar in the seed region and thereby considered a family. This is determined according to the regulation of the mRNA and the functional consequences of the altered expression of the miRNA, in addition to the seed region and the similarity among miRNAs [30].

### 1.3. MiRNAs in Cancer

Recent evidence has confirmed that miRNAs are involved in processes such as cell proliferation, migration, apoptosis, and senescence [31]. For this reason, they have been assigned an important role in the development of various pathologies, including cancer. miRNAs and their characteristics, such as the presence of CpG islands in the promoters of miRNA genes, suggest that methylation has an important effect [32]; if these miRNAs act as tumor suppressor genes, their hypermethylation and subsequent transcriptional silencing would favor tumor progression [33]. An example is the reduction in miR-34 from the hypermethylation of its promoter, which was identified in primary samples of chronic lymphocytic leukemia (CLL), myeloma, and lymphoma [34]. Paradoxically, miRNAs can also regulate the epigenetic machinery [35], and there is evidence suggesting that miR-29b can regulate DNA methylating agents such as DNMT1 and DNMT3 [35]. On the other hand, CpG islands in miRNA promoters are believed to be sites that are more susceptible to affectation by various factors involved in cancerogenesis, since it has been reported that miRNA encoding genes are frequently located in fragile sites or genomic regions typically associated with cancer, called cancer-associated genomic regions (CAGRs) [36].

On the other hand, the regulation of miRNAs is characterized by the fact that their genes have more than one transcription start site [37], which may be located in intronic sequences of some coding genes. Because their transcription is shared with that of the promoter of the coding gene [38], this has allowed a better understanding of the impact of miRNAs on certain functions and their participation in the pathogenesis of cancer [39]. However, many miRNAs are reported to play a bifunctional role as oncogenes and tumor suppressors in different types of cancer [40]. Based on these behaviors, miRNAs have been proposed as diagnostic and prognostic biomarker candidates and potential therapeutic targets in various types of cancer [41], including hematologic malignancies [42]. Recent research has established miRNAs as predictors of chemo- and radio-sensitivity in cancer, and their ability to reverse chemo- or radio-resistance in cancer cells has also been demonstrated [43] by inhibiting or re-expressing them [44]. Reversing the alteration of a specific miRNA by means of synthetic or antagonistic miRNAs was demonstrated to normalize the network of regulatory genes and signaling pathways, revert the phenotype of tumor cells [45], and inhibit tumor cell proliferation and tumor invasion and metastasis [44]. Some of the miRNAs that are involved in cancer are: miR-125b, miR-145, miR-21, and miR-155 in breast cancer [46]; mir-17, -18a, -19a, -20a, -19b-1, and -92a-1 in lymphoma [40]; and miRNA 7 [47].

## 2. MiR-7 in Cancer

After the analysis carried out by the Lagos–Quintana group in 2001 where miRNAs 1 to 33 were studied, some of these were suggested to play an important role in the development of cancer, as well as act in a tissue-specific manner. miR-7 was mentioned for the first time in this publication [10] and was later reported to be encoded in three different loci: 9q21, 15q26, and 19q13, giving rise to three products, pri-miR-7-1, pri-miR-7-2, and pri-miR-7-3, which end in the same sequence as mature miR-7, as shown in Figure 2. Furthermore, it is known to be evolutionarily conserved [48].

miR-7 has a high expression in healthy brain, spleen, and pancreas tissue [49]. Several reports have described this miRNA’s mechanisms of regulation; one of them indicated that the transcription factor HOXD10 is capable of interacting with the promoter region of the putative miR-7 -1 gene to induce the expression of miR-7 [50]. Another transcription factor involved is FOXP3, which induces the expression of miR-7 [51]. Thus, it has also been reported that c-Myc is positively regulates the expression of miR-7 by binding its promoter region [52]. Other proteins involved in pathways such as splicing factors were shown to have regulatory activity toward miR-7, such as the SD2/ASf factor, which interacts with the pri-miR-7 transcript to promote cleavage by Drosha [53]. Mechanisms involved in the immune response were also related to the regulation of miR-7 expression; such is the case of TLR9, the signaling of which increases the expression of HuR, increasing the expression of miR-7 in turn [54]. A regulation of TLR9 by miR-7 was observed in lung cancer, affecting the regulation of the PIK3R3/AKT pathway [55].

On the other hand, a study carried out on various types of cancer indicated that the promotion of the expression of miR-7 together with the reduction in the expression of miR-21, both by viral vectors, resulted in an antitumor effect that encompasses invasion, migration, cell proliferation, and increased survival [56]. In this way, the role of miR-7 in the development of different types of cancer was documented, as shown in Table 1.

### 2.1. MiR-7 in Lung Cancer

Lung cancer is mainly composed of small cell and non-small cell cancer based on its microscopic appearance, with non-small cell cancer (NSCLC) being the most frequent. It is worth mentioning that it is the leading cause of cancer deaths in the United States [57]. In NSCLC, miR-7 was related to the ability to inhibit tumor growth. In a murine model, it was shown that tumor growth and metastasis of lung cancer cells were significantly reduced after inducing the expression of miR-7, which caused a reduction in the signaling of the Akt and ERK in pathways. Furthermore, the downregulation of NDUFA4, a new target of miR-7, was shown to contribute to the same effects of miR-7 expression generated by the TTF-1 promoter [58]. The same effect was demonstrated using drugs that had previously been effective in treating some types of cancer, excluding lung cancer—specifically NSCLC—in results published by Zeng’s group. They evaluated lung cancer cell lines (A549, NCL-H460) and A549 transfected with a mimic or an inhibitor of miR-7 and demonstrated that the compound Breviscapine (BVP), which is a crude extract of erigeron (*Erigeron breviscapuos*) and contains a large number of flavonoids, significantly reduced the growth of lung cancer cell lines and induced apoptosis. In addition, this study proved that cells treated with the drug showed elevated levels of Bax and miR-7. This suggested that the elevation in miR-7 expression promotes the BVP sensitivity of NSCLC cells by suppressing proliferation and promoting apoptosis [59]. Another drug that was shown to increase miR-7 is docetaxel, an antineoplastic drug defined as an antimitotic agent that inhibits the formation of microtubules. In NSCLC cell lines, docetaxel increases the expression of miR-7, and, in turn, this overexpression suppresses proliferation in vitro [60]. The expression of miR-7 also induces inhibition of cancer cell growth, which may be associated with the decrease in the expression of the protein associated with tumor growth CGGBP1, one of the reported targets of miR-7 [61].

Some other targets specifically reported in non-small cell lung cancer are PAX6, a highly conserved transcription factor implicated in NSCLC; miR-7 induces the inhibition of *PAX6* expression, thereby reducing the proliferation potential, migration, and invasion of NSCLC cells via ERK/MAPK [62]. In addition, miR-7 has been reported to inhibit FAK, negative regulation of which is also reflected in inhibiting the activation of the ERK/MAPK pathway, for which both PAX6 and FAK were proposed as new targets for NSCLC treatment [63]. Another reported target is p28α, the overexpression of which has the same effect in NSCLC cell lines as miR-7 downregulation [64]. Thus, the reduced expression of miR-7 was not only found in cell lines, but also in NSCLC tissues [64], while in A549 cells, as well as in NSCLC tissue, miR-7 was shown to inhibit metastasis by NOVA2 targeting [65].

In elucidating the mechanism by which miR-7 acts in tumorigenesis, DNA sequencing has been utilized to determine whether there are mutation sites in the miR-7-2 promoter in lung cancer tissues, and changes such as GC at site -617 and an AG change at site -604 are associated with poor prognosis in cancer patients [66]. The abovementioned mechanisms are illustrated in Figure 3.

Another area under investigation is the role played by miR-7 in resistance to treatment. Studies conducted with paclitaxel (PTX), an antineoplastic drug that inhibits microtubule formation and is the drug of first choice in the treatment of NSCLC, demonstrated that overexpression of miR-7 in NSCLC cell lines increased sensitivity to paclitaxel, suppressed cell proliferation, and increased apoptosis. On the other hand, miR-7 inhibition decreased the anti-proliferative and pro-apoptotic effects of PTX, and miR-7 upregulation enhanced the PTX-sensitivity of NSCLC cells, probably owing to the downregulation of EGFR expression [67]. Data on gefitinib, another drug used in the treatment of NSCLC, which is a tyrosine kinase inhibitor targeting epidermal growth factor receptors or EGFRs, showed that gefitinib treatment coupled with miR-7 significantly decreased the IC50 of gefitinib, inhibited cell growth, increased G0/G1 cell cycle arrest, and increased apoptosis [68]. In a very interesting way and consistent with what was mentioned above, in a study carried out on lung cancer cell lines, it was shown that the lncRNA LNC00240 acts as a miR-7-5p sponge and, as a consequence, induces EGFR overexpression [69].

### 2.2. MiR-7 in Hepatocellular Cancer

Hepatocellular cancer (HCC) or liver cancer is more common in men than in women. Liver cancer can be classified as intrahepatic cholangiocarcinoma or bile duct cancer, angiosarcoma or hemangiosarcoma, or hepatoblastoma. HCC is usually successfully treated with chemotherapy and surgery; however, complications can occur and are more frequent when the cancer originates outside the liver [70].

In HCC, miR-7 is characterized as anti-oncogenic; however, the mechanism is not clear [71]. Studies have been published in which several mechanisms of action were proposed: one of them included the complex formed by nuclear factor 90 (NF90) and nuclear factor 45 (NF45), which acts as a negative regulator of miR-7 processing. The expression of NF90 and NF45 were found to be significantly elevated in primary tumors compared to adjacent non-tumor tissues. After the elimination of NF90 and analysis by miRNA microarray as well as quantitative RT-PCR, results included an elevated expression of mature miR-7 and the decreased expression of pri-miR-7-1 [72]. On the other hand, it has been reported that elements such as voltage-dependent anion channel 1 (VDAC1), which plays an important role in carcinogenesis, play prognostic roles in patients with hepatocellular cancer, and miR-7 was suggested to suppress VDAC1 expression [73]. Another factor related to a poor prognosis is maspin, the expression of which is reduced in patients with hepatitis B virus and correlates with a poor prognosis. Maspin is involved in tumor suppression by promoting cell adhesion and apoptosis, particularly in patients with hepatocellular carcinoma (HCC). Maspin is negatively regulated by hepatitis B virus protein x (HBx). High expressions of miR-7, 107, and 21 were related to the negative regulation of maspin, which was suggested to confer aggressiveness and chemoresistance mediated by HBx in HCC [74].

The involvement of miR-7 was also found in the cell cycle, since cyclin E1 (CCNE1) was identified as an important mediator of the G1/S transition, and reports indicated that this is a direct target of microRNA-7. The overexpression of miR-7 has the effect of silencing CCNE1, thus promoting the tumor suppressor effects attributed to the expression of the CCNE1 oncogene [75]. Finally, it was also reported that miR-7 is capable of negatively regulating SPC24 and, as a consequence, acting as a tumor inhibitor since it suppresses proliferation and migration and promotes apoptosis [76]. The abovementioned mechanisms are illustrated in Figure 4.

### 2.3. MiR-7 in Breast Cancer

Breast cancer manifests itself as abnormal cell growth forming a tumor that can be felt as a lump. The tumor is considered malignant when it is capable of invading surrounding tissues or if it is capable of generating metastases. It is considered almost exclusively a disease of women; however, it can also occur in men, and early detection generally leads to a good prognosis [77].

Various roles for miR-7 have also been reported in breast cancer. Particularly in triple-negative patients, there is usually treatment failure. One of the main characteristics is the elevation in the expression of IL-6, and treatment with antibodies against IL-6 inhibits the migration induced by lapatinib, which is a drug that promotes metastasis in triple-negative breast cancer cells. RAF-1/MAPK, JNK‘s, p38 MAPK and AP-1 activator protein signaling mechanisms are activated in response to lapatinib treatment to induce IL-6 expression. It was reported that miR-7 binds to the 3′UTR region of RAF-1, inhibiting its activity, which leads to the negative regulation of this pathway by miR-7, acting as an antitumor mechanism [78].

miR-7 is also involved in the oncogenic isoform of the HER2 gene, known as HER2Δ16, which is present in 50% of HER2^+^ breast cancer patients. Reports have indicated that this isoform leads to metastasis and resistance to chemotherapy. It was recently shown that HER2Δ16 alters the expression of some miRNAs. Using an array comparison of miRNA expression profiles of the MCF/7 and MCF7/HER2Δ16 cell lines, 16 miRNAs were found to be significantly altered by two or more folds. miR-7 showed a greater increase, at 4.8-fold in the MCF7/HER2Δ16 lines. The restoration of miR-7 expression promotes G1 cell cycle arrest and reduces colony formation and cell migration to levels similar to MCF7, while the suppression of miR-7 in MCF7 results in promotion of colony formation, but not in cell migration. The mechanism involved includes EGFR and Src Kinase, which represent targets for therapeutic intervention and refraction and metastasis of breast cancer with HER2Δ16 [79].

Another signaling pathway in which miR-7 participates is WISP2 (WNT1-inducible-signaling pathway protein 2); silencing WISP2 signaling in the MCF7 cells of human breast adenocarcinoma impairs cytotoxic T-lymphocyte (CTL)-mediated cell death through a mechanism that involves the induction of Kruppel-like factor-4 (KLF-4) and miR-7 decrease. Using a WISP2 inhibitor results in a significant reversal of the epithelial-to-mesenchymal transition. KLF4 expression was found to be related to the inhibition of miR-7, which is responsible for CTL-mediated lysis. WISP2 has a role in tumor susceptibility through EMT via the TGF-β signaling pathway, KLF4 expression, and miR-7 inhibition, thus proposing WISP2 as an activator of CTL-induced death and suggesting that the loss of its function promotes immune surveillance evasion and tumor promotion [80]. On the other hand, in stem-type breast cancer cell lines with metastatic capacity to the brain, a profile of miRNAs was made that showed a reduced expression of miR-7, which promotes the expression of KLF4 factor in induced pluripotent stem cells. It was reported that miR-7 expression can decrease KLF4 expression and suppress breast cancer cell metastasis [81].

Another report indicated that in breast cancer cell lines and through qRT-PCR, a high expression of miR-7 is directly related to a better response to chemotherapy with paclitaxel/carboplatin through the suppression of MRP1 and BCL2 [82]. The mechanisms mentioned above are illustrated in Figure 5.

### 2.4. Role of MiR-7 in Gliomas

Gliomas are among the ten most common causes of death related to cancer. Although there is no test to detect it early, timely diagnosis and treatment improve the outcome. Recent progress has been made in the scientific understanding and in methods of diagnosis and treatment [83]. Alterations in energy metabolism were observed in gliomas involving Akt signaling in aerobic glycolysis programs; however, the mechanisms regulating aerobic glycolysis and Akt activity are not known. One of the proposed mechanisms includes the regulation of IGF-1R, which is part of the signaling pathway upstream of Akt. The increase in the expression of miR-7 was reported to inhibit colony formation and the glucose metabolic capacity of glioma cells, an effect similar to that produced by the “knockdown” of IGF-R, which is consistent with data indicating that miR-7 is capable of binding to the 3′UTR region of IGF-R, causing its decrease. Expression data of miRNAs downloaded from the Genomic Atlas of Cancer database were analyzed [84]. Thus, the expression profiles of 20 mature miRNAs were also studied in patient biopsies with the finding that miR-7 is less expressed in tumors compared to healthy control samples [85]. In addition to the Akt pathway, miR-7 is involved in the regulation of the RAF/ampk (MEK) and ERK pathways, two of which are initiated by EGFR, which in glioblastoma is reported to be a regulator of cell proliferation, survival, migration, and invasion. An increase in the expression of miR-7 decreases the expression of PI3K, phosphorylation of AKT, RAF-1, and phosphorylated MEK1/2, and causes a slight reduction in the expression of EGFR. An induced expression of PI3K or RAF-1 reverses the effects of miR-7. Therefore, miR-7 was proposed as a potential tumor suppressor because it is capable of regulating several related oncogenes in the EGFR pathway [86]. Additionally, the overexpression of miR-7-5p was found capable of inhibiting SATB1 (SATB homeobox 1) and reversing the effects of promoting cell migration and invasion in glioblastoma cells [87]. Another target of miR-7-5p in glioblastoma is YY1 (Ying Yang 1), the silencing of which has direct effects on the reduction in the chemoresistance of cells to temozolomide [88].

Furthermore, the first trials using miRNA as part of a therapy were performed with promising results; the administration of miR-7 encapsulated in cationic liposomes in xenografts arrested growth, and metastatic nodules decreased the effectiveness in a sequence-specific manner. miR-7 has been applied in the treatment of glioma, and the results have shown it to be a promising potential anti-tumor and anti-metastatic treatment for human glioma [89]. All the mechanisms mentioned above are illustrated in Figure 6.

### 2.5. Role of MiR-7 in Colorectal Cancer

Colorectal cancer (CRC) can occur in the colon or rectum, hence the name; most cases of this type of cancer begin with a growth called a polyp in the inner lining of the colon. Adenocarcinomas represent about 95% of colorectal cancers [90]. Colorectal cancer studies conducted on 196 malignant and 14 normal tissues, analyzing the expression of miR-7 levels with real-time PCR, found that miR-7 was significantly elevated in colorectal cancer compared to normal colorectal mucosa. From this study, it was determined that a high expression of miR-7 is significantly associated with poor survival in patients with CRC (*p* = 0.01) [91]. In addition, it was determined by in vitro assays that EGFR and RAF-1 are targets of miR-7 [92]. Additionally, it was reported in colorectal cancer that the 3′UTR region of YY1 had binding sites for miR-7, which was confirmed by reporter plasmid assays, and that miR-7 was downregulated in colorectal cancer cell lines and in the ectopic expression results on the suppression of cancer cell proliferation, apoptosis induction and cell cycle arrest. In parallel, the “knockdown” of YY1 suppresses cell proliferation and induces apoptosis, indicating the inverse functions of miR-7 and YY1 in CRC cell lines, where YY1 promotes CRC proliferation and miR-7 inhibits it by inhibiting YY1 expression. Based on this, miR-7 was proposed as a tumor suppressor by suppressing YY1 in its oncogenic role. YY1 can promote cell growth by inhibiting p53 and promoting the Wnt signaling pathway [93]. Therefore, the precursor of miR-7 was proposed as a possible therapy. In contrast, in another study carried out on CRC tissues and cells, it was determined that a low expression of miR-7-5p in CRC tissue predicts a low 5-year survival, while in cells, it was determined that the overexpression of this microRNA is directly related to the inhibition of cell proliferation and migration. This represents a controversial finding on the expression of miR-7 in CRC, which should be addressed and based on the targets of miR-7 in this context. Additionally, it was reported that KLF4, a target of miR-7, acts as an oncogene in colorectal cancer, so the inhibition of KLF4 expression by miR-7 is a mechanism and a potential molecular target for the treatment of colorectal cancer [94]. TRIP6 is another target of miR-7 that was related to the regulation of metastasis and inhibition of cell proliferation; however, in colorectal cancer cells, it was reported that miR-7 is decreased as a consequence of hypermethylation of TRIP6, while the restoration of miR-7 expression has a direct impact on the inhibition of cell proliferation and migration [95]. Interestingly, it was reported that, in biopsies of patients diagnosed with colorectal cancer, the expression of miR-7-5p together with miR-10a-5p can serve as a marker of poor prognosis [96]. The mechanisms mentioned above are illustrated in Figure 7.

### 2.6. Role of MiR-7 in Prostate Cancer

The prostate is a gland that only men have, and the abnormal growth of the cells generates prostate cancer, which alters the gland’s growth. The prostate is normally compared to a walnut in regard to size. Prostate cancer is more common in men of 40 years or older [97].

miR-7 has been described as a tumor suppressor capable of annulling the stemness of prostate cancer cells, inhibiting tumorigenesis by suppressing KLF4. The possible mechanism is the regulation of the KLF4/PI3K/Akt/p21 pathways [98]. Studies were conducted on PC3 and LnCap prostate cancer cell lines, where it was found by means of an expression profile that miR-7 is increased in PC3 (11.3 times increased, *p* = 0.012). However, in patients with a high expression of miR-7 in peripheral blood, a significantly decreased survival rate was observed. Therefore, miR-7 is proposed as a potential biomarker in the prognosis of prostate cancer [99].

### 2.7. MiR-7 in Oral Cancer

Oral cancer is the neoplasm that occurs in any of the regions of the oral cavity; 90% of these correspond to squamous cell carcinoma, and others include adenocarcinoma derived from the minor salivary gland, sarcoma, malignant lymphoma, or metastasis from other malignancies [100]. Recent studies on 85 samples of esophageal squamous cell carcinoma as well as a database analysis revealed that there is a higher expression of miR-7, compared to normal tissue. Thus, low levels of miR-7 are associated with poor prognosis. A multivariate analysis showed that low miR-7 expression was an independent predictor of poor survival. On the other hand, in vitro assays showed that the miR-7 precursor suppresses the proliferation of esophageal squamous cell carcinoma, thus proposing miR-7 as a prognostic marker and therapeutic target [101]. In a study, blood sera from 105 patients with esophageal carcinoma were evaluated, and the expression of miR-7 in patients with esophageal squamous cell carcinoma was 4.74 times lower than that of healthy subjects, again indicating its potential as a diagnostic factor. Its expression in patients who responded to chemoradiotherapy treatment (CRT) was 2.34 times higher than in those who did not respond, indicating its potential as a predictive biomarker for response [102].

### 2.8. MiR-7 in Thyroid Cancer

Thyroid cancer occurs when there is an uncontrolled growth of the gland; it originates in thyroid follicular cells and can be classified as papillary, follicular, or Hurtle cell cancer, the latter having the worst prognosis, while the first two have a good prognosis and generally respond well to treatment [103]. Several reports on thyroid cancer have established the regulation of signaling pathways by miR-7; through reporter plasmid assays, it was shown that miR-7 can bind to the 3′UTR region of CKS2. Using Western blot, it was determined that microRNA-7 negatively regulates the expression of the CKS2 protein and its downstream genes, such as Cyclin B1 and cdk1. A knockdown model of CKS2 by siRNA in TPC1 and K1 cells suppresses cell proliferation, cell migration and invasion [104]. Thus, it was also found that miR-7 is capable of regulating PAK1 expression at the protein level, but not at the RNA level. Furthermore, the restoration of miR-7 expression as well as PAK21 knockdown were shown to be similarly related to a suppression of thyroid cancer cell proliferation, migration, and invasion [105]. The mechanisms mentioned above are illustrated in Figure 8.

### 2.9. Role of MiR-7 in Melanoma

Melanoma is known as a cancer originating in melanocytes. Melanomas have a high production of melanin, which produces a brown or black coloration. It usually occurs in the extremities, trunk, and chest. Melanoma detected in its early stages usually has a good prognosis; however, in advanced stages, it has a high probability of invading other tissues. It is divided into skin cancer and basal and squamous cell cancer, the latter having the best prognosis [106].

Studies have been conducted specifically on the A375 and A375 melanoma cell lines, which are resistant to vemurafenib (VemR), a drug that targets mutated BRAF-type protein kinases. Low miR-7 expression was found through miRNA microarray analysis, and by restoring miR-7 expression, Vem resistance was reversed. In addition, miR-7 mimic-induced expression was shown to decrease the expression of EGFR, IGF-1R, and CRAF, which are regularly overexpressed in VemR cells and decreased tumor growth [107]. Thus, the expression of miR-7-5p was decreased compared to primary melanoma, and its increased expression reduced the viability of tumor cells and the formation of colonies and induced cell cycle arrest. It is also a potent inhibitor of melanoma growth and metastasis in part by inactivating RelA/NFκB signaling [108].

### 2.10. MiR-7 in Cervical Cancer

Worldwide, cervical cancer is a main cause of death in women. It was linked to the presence of human papillomavirus, and infection by human papillomavirus was proven to be a requirement for the development of the disease. Cytological studies such as Pap smears are essential to early detection and thus to the efficacy of treatment [109]. In cervical cancer, there is a low expression of miR-7, especially in metastatic tumors; however, the ectopic expression of said miRNA significantly inhibits FAK in HeLa and C33A cells, a protein involved in the regulation of cell proliferation and survival at the transcriptional and protein levels, which correlates with the negative correlation of miR-7 and FAK in cervical cancer tissues [30]. It was also reported to promote apoptosis and loss of cell viability in these cell lines by inhibiting one of its targets, the XIAP protein [110].

### 2.11. Role of MiR-7 in Other Types of Cancer

Regarding other types of cancer, in pancreatic carcinoma miR-7 is involved in the regulation of the ILF2 oncogene in the epithelial–mesenchymal transition in PANC-1 cells, functioning as a supposed pro-tumor factor. Specifically, it was shown that curcumin promotes miR-7 expression, which, in turn, suppresses ILF2 expression at the mRNA and protein levels [111]. In adrenocortical carcinoma, the systemic administration of miR-7 through a novel therapy consisting of delivery vesicles (EGFREDVTM nanocells) allowed us to observe that miR-7 inhibited cell proliferation in vitro and induced cell cycle arrest in G1; it also reduced the growth of transplanted tumors from both cell lines and primary cultures of adrenocortical carcinoma. The proposed mechanism of action is based on the fact that miR-7 acts on the proto-oncogene RAF-1 and mTOR, and miR-7 therapy inhibits cyclin-dependent kinase 1 (CDK1), which was increased in samples from patients with adrenocortical carcinoma, inversely correlated with miR-7 expression [47].

In gastric cancer, “genome-wide screenings” were carried out where RelA and FOS proteins were identified as targets of miR-7; repressing the expression of these proteins in turn prevented proliferation and tumorigenesis in gastric cancer cells. Conversely, low miR-7 expression correlated with high RELA and FOS expression and poor survival in GC patients. miR-7 indirectly regulates RELA activation by targeting the IκB kinase IKKε. In addition, IKKε and RELA can repress miR-7 transcription, which forms a feedback loop between miR-7 and nuclear factor κB (NF-κB) signaling. The downregulation of miR-7 results in aberrant activation of NF-κB signaling by *Helicobacter pylori* infection [112].

In ovarian cancer, EGFR, which promotes tumor proliferation and migration, was reported to be a direct and functional target of miR-7. miR-7 expression was shown to be decreased in highly metastatic epithelial ovarian cancer (EOC) cell lines and metastatic tissues, whereas EGFR expression was positively correlated with metastasis in both EOC patients and cell lines. miR-7 inhibits tumor metastasis and reverses the epithelial–mesenchymal transition through AKT/ERK1/2 inactivation by targeting EGFR in epithelial ovarian cancer [113].

miR-7 expression was demonstrated in renal cell carcinoma tissues, and the negative regulation of miR-7 inhibited cell migration in vitro, suppressed cell proliferation, and induced apoptosis of renal cancer cells [114]. The mechanisms mentioned above are illustrated in Figure 9.

### 2.12. MiR-7 in Hematologic Malignancies

In hematological malignancies, genes have been found that are directly related to the generation of malignancy. One of them is TET2 (Ten-Eleven-Translocation 2), which is considered to be an important tumor suppressor that is frequently mutated. Through the analysis of 3‘UTR activity, miRNAs that inhibit TET2 expression were identified. Among them, a high expression of miR-7 was found in acute myeloid leukemia, and it was shown that miRNAs have potential for diagnosis [115]. On the other hand, in children with acute lymphoblastic leukemia, an abnormal expression of miRNAs was identified by means of miRNA expression profiles and confirmed by qRT-PCR, in relation to children with thrombocytopenia as a control. In fact, this was proposed through a correlation with the expression of miRNAs, finding that miR-7 together with other miRNAs could be involved in early relapse, particularly when miR-7, miRNA-216, and let-7i are highly expressed [116].

On the other hand, it is known that blood microvesicles can contain various components, including miRNAs. Analyses have been carried out in which the expression of miRNAs was studied to establish standards that compare those from leukemia-derived microvesicles and normal cells. Analyses of 888 miRNAs were performed, and most of the miRNAs that are deregulated were found to be involved in signaling pathways associated with leukemia, particularly MAPK and p53. Considering the translocation present in leukemia, which involves chromosome 9, the functions of 12 miRNAs were examined, among them miR-7, which are expressed in K562 cell vesicles and in K562 cells and are proposed as potential therapeutic targets [117]. Another mutation in leukemias, particularly chronic lymphocytic leukemia, is the p53 mutation. It was shown that there are several p53-dependent miRNAs induced in response to DNA damage in CLL, proposing that the miRNoma for the identification of the dependent DNA damage response machinery stems from p53 [118]. In ALL (acute lymphoblastic leukemia) of T lymphocytes, it was reported that the overexpression of miR-7 is capable of suppressing TAL1 levels and reducing growth, mobility, and migration, as well as promoting the induction of apoptosis, allowing it to play a potential role as tumor suppressor [119]

Additionally, it was reported that in lymphomas of the anaplastic lymphoma kinase-positive anaplastic large cell lymphomas subtype, miR-7 -5p inhibits RAF1, similar to using siRNAs or vemurafenib. The latter is a chemical inhibitor of BRAF, causing an increase in autophagy and potentiating the toxicity of crizotinib, a tyrosine kinase inhibitor [120]. In MALT lymphoma patients, there is decreased expression of miR-7 and miR-153 compared to patients with gastritis [121], which suggested, based on what was previously shown, the crucial participation of miR-7 in the inhibition of lymphomagenesis. Interestingly, it was reported that a decrease in the expression of miR-7 carried out by c-MYC resulted in an overexpression of Fas-L in M1 macrophages, thus promoting greater apoptosis in tumor stromal macrophages (tumor stromal macrophages). This low expression of miR-7 was also related to poor clinical outcomes and with the evolution of low-grade follicular lymphoma to aggressive follicular lymphoma [122].

In addition, tumors from 16 lymphoma patients were scanned to analyze 851 human miRNAs, finding a significant difference in 133 miRNAs (>2-fold; *p* < 0.05) between lymphoma and follicular hyperplasia. In particular, it was determined that miR-7 is increased in follicular lymphoma but decreased in a subgroup of cases. Furthermore, high expression was associated with improved response to chemotherapy [123]. Recently, our research group reported the participation of miR-7 in the regulation of migration and chemoresistance in NHL cell lines, through the regulation of YY1 and KLF4. Additionally, in the analysis of expression in biopsies of patients diagnosed with NHL (follicular lymphoma and diffuse large B-cell lymphoma), we showed that there is a negative correlation between the expression of both transcription factors YY1 and KLF4 vs. miR-7, which is indicative of miR-7 regulation of these transcription factors. Therefore, as a whole, the results described the participation of miR-7 with KLF4 and YY1 in the context of non-Hodgkin’s lymphoma [124], which was consistent with what was reported in other types of cancer (Table 1). The mechanisms mentioned above are illustrated in Figure 10.

## 3. MiR-7 and Its Clinical and Therapeutic Potential in Cancer

As mentioned in this review and in others, miRNAs have potential as biomarkers and as therapeutic agents or targets. In particular, miR-7 has been shown to act mainly as a tumor suppressor, inhibiting a large number of oncogenic targets, owing to the fact that reduced levels of this miRNA were involved in the development [48]. It has an effect on several of the hallmarks of cancer, such as initiation of malignancy, proliferation, migration, invasion, survival and death [125], resistance to therapies, and its progression both in vitro and in vivo (Table 2). However, miR-7 has also been reported to promote cancer [126].

Additionally, we performed a gene expression analysis of the data available in the GEO database (GSE10846) and strictly delimited the clinical information reported by A. Alizadeh et al. [127]. A Kaplan–Meyer survival analysis of miR-7 expression was conducted relative to follow-up. In relation to the average of the expression, we divided the expression of miR-7 into high and low, determining that the high expression of miR-7 was related to a lower percentage of survival in NHL (Figure 11A). On the other hand, we analyzed miR-7 expression in pediatric acute lymphoblastic leukemia, using the database available in GEO (GSE23024) [128]. We found that there is a higher expression of miR-7 in those patients with resistance to prednisone (Figure 11B) but not to daunorubicin or vincristine. In addition, we analyzed the GEO database (GSE124489) [129], and miR-7 expression in multiple myeloma indicated a trend of higher miR-7 expression in newly diagnosed patients compared to healthy donors (Figure 11C). As we previously reported [124], the expression of miR-7 was decreased in DLBCL dead patients compared with the living patients in the data set GSE31312 [130] (Figure 11D) (figure analysis was published previously [124]). Taken together, these data suggested that miR-7 plays a fundamental role in the development of hematological malignancies.

For all these reasons, miR-7 has been proposed as a potential strategy for cancer therapy owing to the significant progress made in understanding the molecular mechanisms of miR-7 and the regulatory networks it affects. There is sufficient evidence to suggest that miRNA replacement therapy may be a new alternative to the treatment of various diseases, including specific human cancers [48].

## 4. Conclusions

miR-7 is a miRNA with an important role in various processes related to cancer. This was confirmed in several types of malignancies where miR-7 participated in the regulation of signaling pathways, as well as in the regulation of transcription factors that serve as biomarkers and therapeutic targets. miR-7 has been proposed as a biomarker as well as a potential therapeutic target and tool for the treatment of various types of cancer.

## Figures and Tables

**Figure 1 ijms-23-09091-f001:**
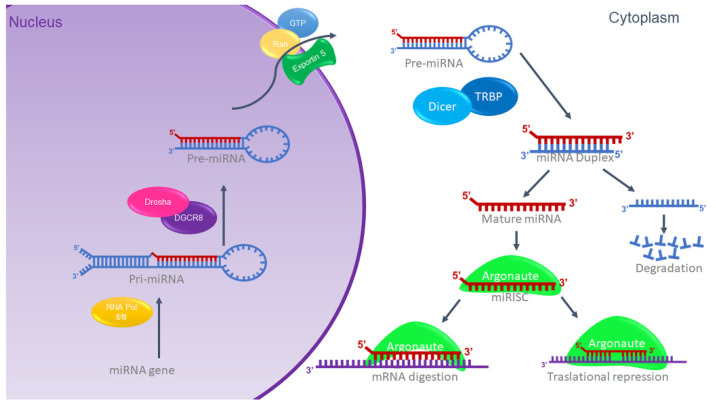
miRNA biogenesis. Schematic representation of canonic miRNA biogenesis. Modified from G. Hutvagner, 2002 [26].

**Figure 2 ijms-23-09091-f002:**
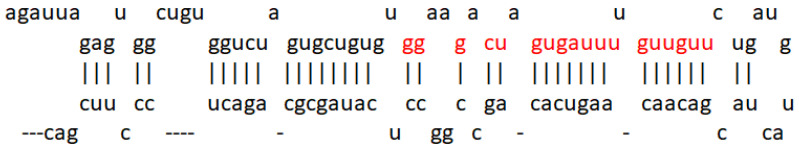
miR-7 sequence. Representation of the microRNA-7 sequence obtained from miRDB (http://mirdb.org/cgi-bin/mature_mir.cgi?name=hsa-miR-7-5p accessed on 1 June 2022). The seed region is highlighted in red.

**Figure 3 ijms-23-09091-f003:**
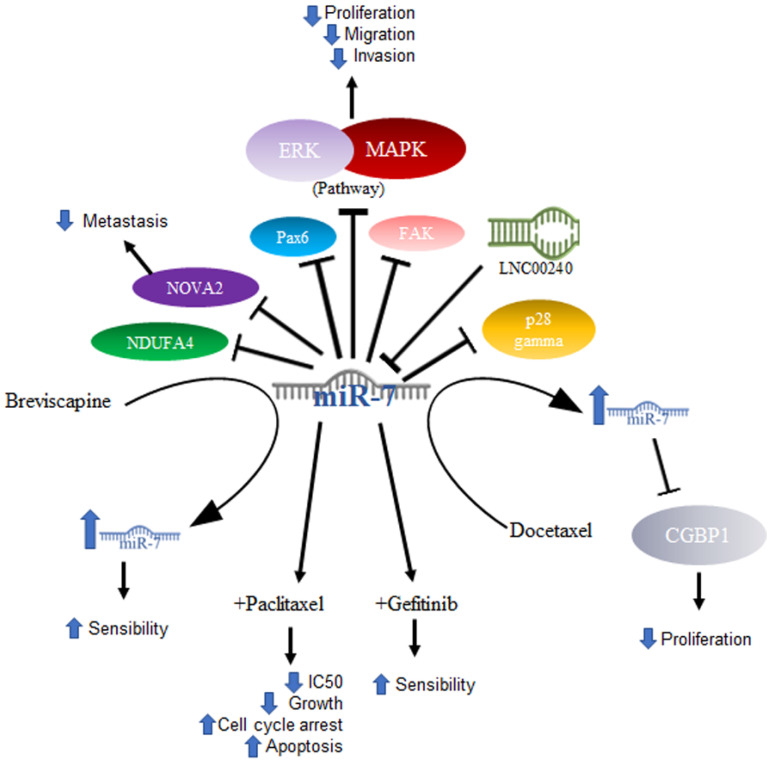
Role of miR-7 in lung cancer. Schematic representation of protein interactions, roles, and treatment with miR-7 and its biological consequences in lung cancer.

**Figure 4 ijms-23-09091-f004:**
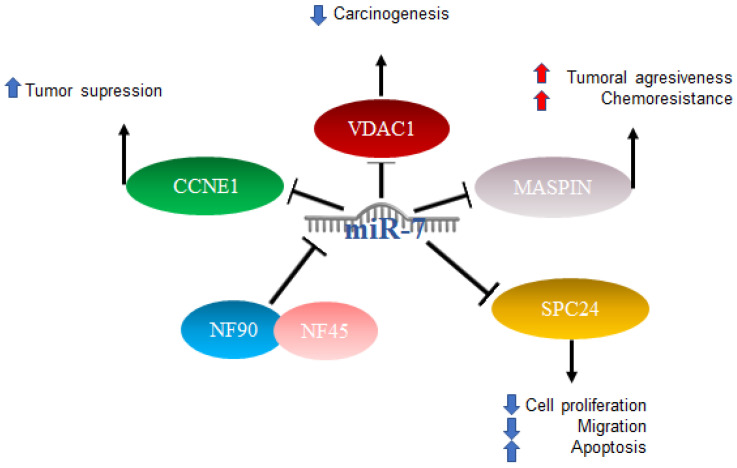
Role of miR-7 in hepatocellular cancer. Schematic representation of protein interactions and roles of miR-7 and its biological consequences in hepatocellular cancer.

**Figure 5 ijms-23-09091-f005:**
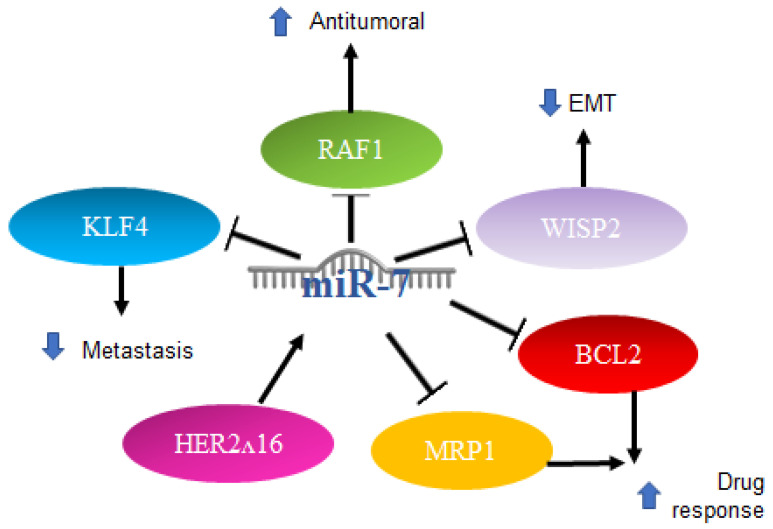
Role of miR-7 in breast cancer. Schematic representation of protein interactions and roles of miR-7 and its biological consequences in breast cancer.

**Figure 6 ijms-23-09091-f006:**
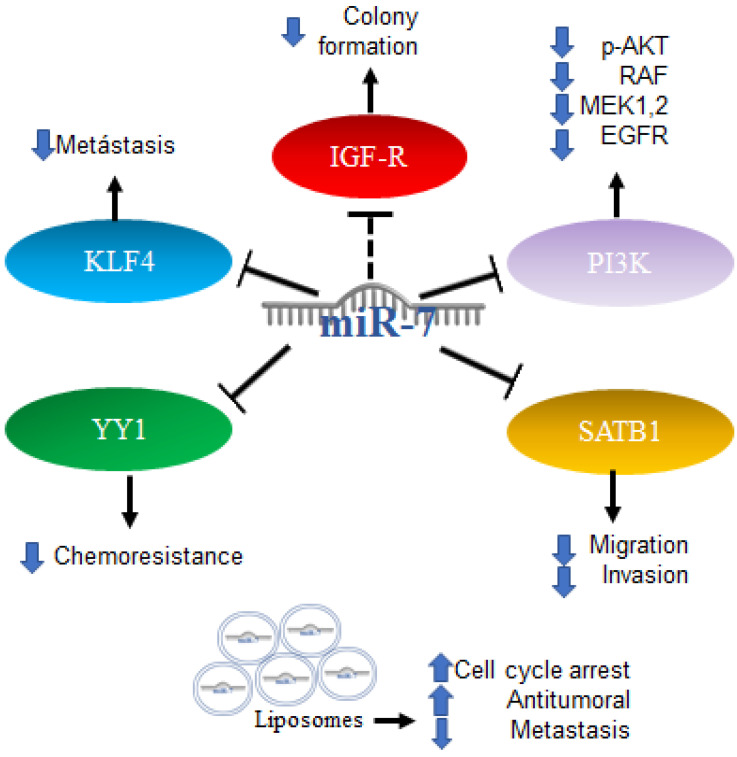
Role of miR-7 in gliomas. Schematic representation of protein interactions, roles and treatment with miR-7 and its biological consequences in gliomas.

**Figure 7 ijms-23-09091-f007:**
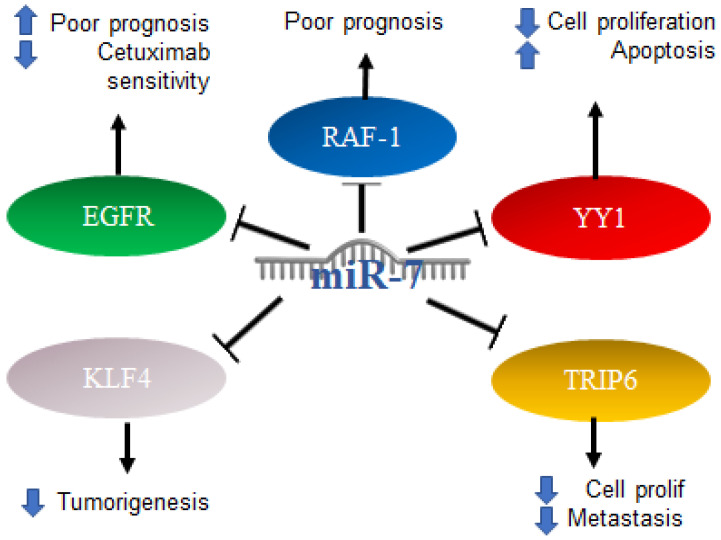
Role of miR-7 in colorectal cancer. Schematic representation of protein interactions with miR-7 and its biological consequences in colorectal cancer.

**Figure 8 ijms-23-09091-f008:**
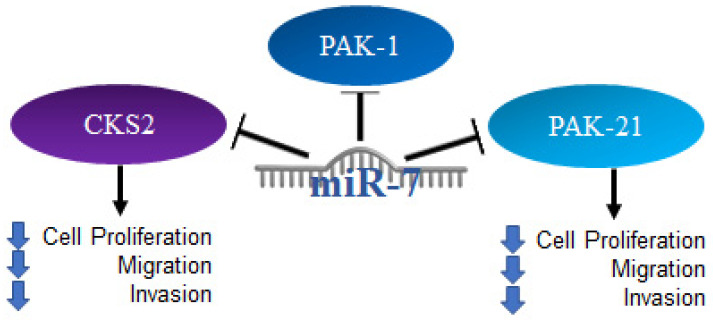
Role of miR-7 in thyroid cancer. Schematic representation of protein interactions with miR-7 and its biological consequences in thyroid cancer.

**Figure 9 ijms-23-09091-f009:**
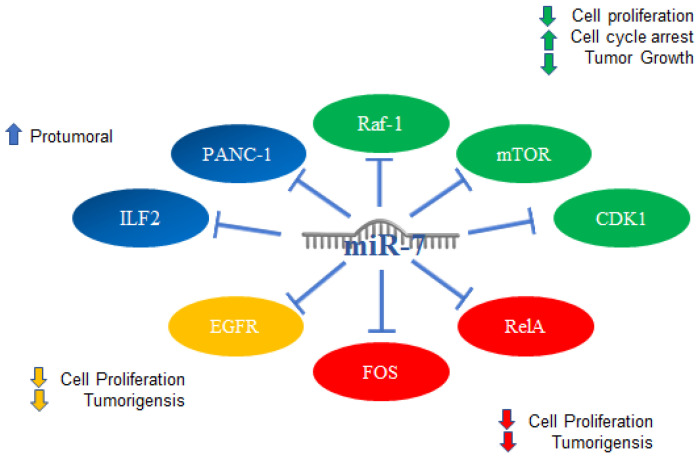
miR-7 in other types of cancer. Schematic representation of protein interactions with miR-7 and its biological consequences in several type of cancer, (blue) pancreatic, (green) adrenocortical cancer, (red) gastric, and (yellow) ovarian.

**Figure 10 ijms-23-09091-f010:**
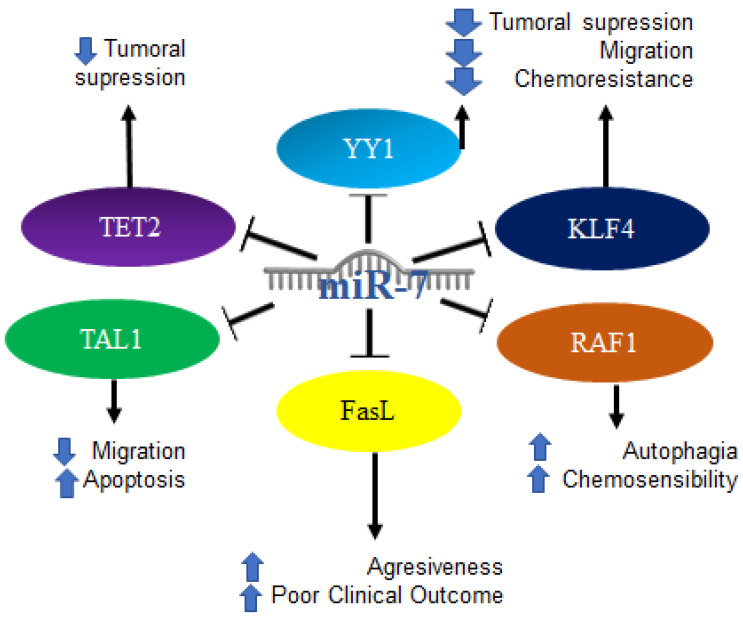
Role of miR-7 in hematological malignancies. Schematic representation of protein interactions with miR-7 and its biological consequences in the context of hematological malignancies.

**Figure 11 ijms-23-09091-f011:**
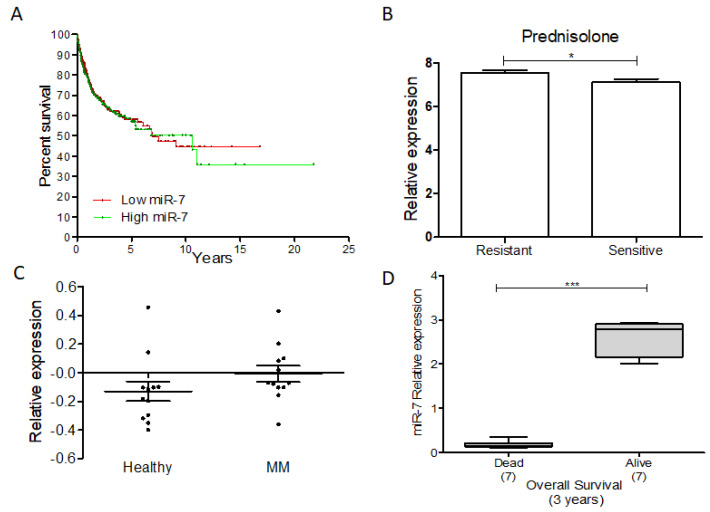
miR-7 and its clinical and therapeutic potential in cancer. miR-7 as a potential biomarker in some hematological malignances. (**A**) GEO database analysis expression of miR-7 in NHL. (**B**) GEO database analysis expression of miR-7 in pediatric-ALL; Student’s *t* test was used. (**C**) GEO analysis of miR-7 expression on MM; Student’s *t* test was used. (**D**) GEO database analysis expression of miR-7 in DLBCL NHL; Student’s *t* test was used (* *p* < 0.01, *** *p* < 0.001).

**Table 1 ijms-23-09091-t001:** miR-7 targets and functions of target genes in cancer. Taken and modified from “microRNA-7: A tumor suppressor miRNA with therapeutic potential” [48].

Target	Function	Type of Cancer	References
EGFR	Promotes cell survival, proliferation, tumorigenesis, resistance to therapeutic targets and radiotherapy	Head and neck, Glioblastoma multiforme, cervical, lung, breast, and prostate	[2,3,4,5,6]
RAF1	Promotes cell survival	Lung, breast, and HNC	[3,5,7]
PAK1	Promotes cell survival, proliferation, cell motility, invasion, growth, and tumorigenesis	Breast, squamous cell carcinoma of the tongue, Schwannoma	[6,8,9]
IRS-1	Promotes proliferation	Glioblastoma multiforme, breast and cervical	[8,10]
IRS-2	Promotes cell survival, proliferation, motility, and invasion	Glioblastoma multiforme, lung, breast, prostate, squamous cell carcinoma of the tongue, and Schwannoma melanoma	[3,8,9,10,11]
ACK1	Promotes cell proliferation, and tumorigenesis	Schwannoma	[6]
PI3KCD	Promotes cell survival, proliferation, tumorigenesis, and metastasis	Hepatocellular carcinoma	[12]
mTOR	Promotes cell survival, proliferation, tumorigenesis, and metastasis	Hepatocellular carcinoma	[12]
P70S6K	Promotes cell survival, proliferation, tumorigenesis, and metastasis	Hepatocellular carcinoma	[12]
BCL-2	Promotes resistance to apoptosis, proliferation, and tumorigenesis	Lung	[13]
XIAP	Promotes cell survival and proliferation	Cervical	[14]
YY1	Promotes cell survival, proliferation, and tumorigenesis	Colorectal	[15]
CCNE1	Promotes cell survival	Hepatocellular carcinoma	[15,16]
PA28γ gamma	Promotes cell survival, proliferation, and tumorigenesis	Lung	[17]
FAK	Promotes cell proliferation, cell survival, tumorigenesis, and cell mobility and regulates EMT	Glioblastoma multiforme and breast	[18,19]
KLF4	Promotes metastasis and self-renewal of stem-type cancer cells	Breast	[20]
IGF1R	Promotes cell survival, proliferation, migration, invasion, and metastasis	Squamous cell carcinoma of the tongue, gastric	[9,21]
MRP1	Promotes resistance to chemotherapy	Breast	[22]
ERF	Represses checkpoints in the cell cycle	Lung	[23]

**Table 2 ijms-23-09091-t002:** miR-7 roles in cancer.

Type of Cancer	MiR-7 Expression Levels	Role	Reference
Lung	Low	Inhibits tumor growth and metastasis Promotes chemosensitivity	[24,25]
Hepatocellular	Low	Tumor suppression by CCNE1 inhibition	[26]
Breast	High	Cell cycle promotion	[27]
Gliomas	Low	Inhibits oncogenes Suppresses metastases	[20,28]
Colorectal	High	Tumor suppressor through YY1 inhibition	[92]
Prostate	High	Tumor suppression by inhibition of KLF4	[30]
Oral	Low	Proliferation suppression	[31]
Thyroid	ND	Suppresses cell proliferation, migration and invasion	[32]
Melanoma	Low	Reverses resistance, decreases tumor growth	[33,34]
Cervix	Low	Promotes apoptosis, decreases cell viability.	[14,35]
Pancreatic	ND	Suppression of the epithelial–mesenchymal transition	[36]
Adrenocortical	Low	Reduction in cell proliferation	[37]
Gastric	ND	Prevents cell proliferation and tumorigenesis	[38]
Ovary	Low	Inhibits tumor metastasis and reverses epithelial–mesenchymal transition	[39]
Follicular lymphoma	High	Associated with better response to chemotherapy	[40]
Lymphoblastic leukemia	High	Related to early relapse.	[41]

## Data Availability

All the datasets used in this review were obtained from the GEO DATABASE, available at https://www.ncbi.nlm.nih.gov/geo/ Last accession to the database was 1 June 2022 for all; GSE10846 Link: https://www.ncbi.nlm.nih.gov/geo/query/acc.cgi?acc=GSE10846; GSE23024 Link: https://www.ncbi.nlm.nih.gov/geo/query/acc.cgi?acc=GSE23024; GSE124489 Link: https://www.ncbi.nlm.nih.gov/geo/query/acc.cgi?acc=GSE124489; GSE31312 Link: https://www.ncbi.nlm.nih.gov/geo/query/acc.cgi?acc=gse31312.

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
