# Peer review of "Role of MicroRNA-7 (MiR-7) in Cancer Physiopathology"

_ijms, 2022, doi:10.3390/ijms23169091_

Round 1

Reviewer 1 Report

This manuscript attempts to give an overview of the role of miR-7 in cancer but falls short as it does not sufficiently present the reader with a critical assessment of the available literature and lacks a clear perspective. In addition, the authors should pay close attention to style which should be clear and unambiguous.  

Major Comments:

1)      Page 7, line 206 – 219 – Is EGFR targeted by miR-7-5p, it is according to table 1? How then can miR-7 inhibition decrease sensitivity to paclitaxel through decreased EGFR expression? Does this reflect a controversy in the literature?

 2)      Page 12, line 355 – What is exactly meant by the statement: “..indicating the inverse functions of miR-7.”? Please explain more clearly. How is it that miR-7 overexpression has been proposed as therapy whereas miR-7 is already found elevated in colorectal cancer (line 348-349)? It could be that findings on miR-7 and colorectal cancer from different research groups do not agree, however, as a reader one likes to know what is most likely true.

 3)      Page 18, line 568 – Why was the average used to discriminate high miR-7 from low expressors? Usually, the median is chosen to distinguish.

 4)      Figure 11 – Which statistics were used, please be clear on this.

 Minor Comments:

1)      Carefully check English grammar and wording throughout the manuscript. Some sentences are quite long and difficult to follow. Also check the use and spelling of gene names.

2)      Page 1, line 14 – 15 – “…a fundamental part of the strategy in cancer therapy….” This part of the sentence is not very clear, please rephrase.

 3)      Page 2, line 46 – “Caenorrabdihtis” is misspelled, please check and correct.

 4)      Page 2, line 51 – It should be “C. elegans”, please correct.

 5)      Page 2, line 53 – 57 – Please check and rephrase this paragraph making it more comprehensible.

6)      Figure 1 – Please give the proper reference from which the figure is derived.

7)      Page 3, line 85 – Note that there are currently far more than 1900 human microRNAs are listed in miRbase. Please check and correct.

 8)      Sometimes miRNAs are referred to as “microRNAs” sometimes as “miRNAs”. Please be consistent.

 9)      Page 3, line 102 – 106 – This sentence is not clear, what do the author try to convey? Please check and rephrase.

 10)   Page 6, line 175 – Do you mean by “Erigien Breviscapuos” the plant “Erigeron breviscapus”? Please check and correct.

 11)   Page 6, line 187 – Should it be “pax6” (no capitals, indicative of the murine protein) or “PAX6” (capitals signifying the human protein). Is it Pax6 or Pax-6? Please be consistent.

 12)   Page 6, line 193 – How does the sentence “for the treatment of NSCLC [63].”  fit in?

 13)   Page 6, line 199 – Which miR-7 gene was studied? (miR-7-1, miR-7-2 or miR-7-3)

 14)   Figure 3 – “Placlitaxel” should be “Paclitaxel”, please change.

 15)   Page 7, line 214 – “Gefintinib” should be “Gefitinib”

 16)   The heading for section 2.3 (Role of miR-7 in breast cancer) is missing, please amend.

 17)   Page 12, line 378 – It is not correct to compare prostate cancer growth to a walnut. What the authors intended to state is probably that the prostate’s size normally compares to a walnut. Please check and correct.

 18)   Page 13, line 398 – 403 – In which tissue or body fluid is miR-7 measured in the 105 patients with esophageal carcinoma?  To which treatment did the patients respond?

 19)   Page 13, line 416 – “PAK21 know down” what is meant, please check and correct.

20)   Page 14, line 441-445 – This sentence does not read well, please check and correct.

 21)   Page 15, line 472-473 – The “κ” (kappa) symbol is missing from NF- κB, please check and correct.

 22)   Page 16, line 548 – 549 – This Table heading is misplaced, please correct.

Author Response

See atached file

Reviewer 2 Report

Overall, this a well written review focusing on a microRNA , miR-7 which has been described as an imprortant factor in the development of cancer, due to its role as a tumor suppressor, whih regulates a broad spectrum of genes genes involved in the development and progression of cancer. This review is important since recent data propose miR-7 as a prognostic biomarker in cancer and as a fundamental part of the strategy in cancer therapy. In this work, the role played by miR-7 in various types of cancer has been thoroughly  reviewed, showing its regulation, its direct targets and its effects, as well as the possible relationship it has with the clinical outcome of cancer patients. Thu,s this is an excellent review worthy enough for its publication

Author Response

The authors appreciate the reviewer's comments.

Round 2

Reviewer 1 Report

The authors did address my initial comments well, there are a few remaining issues:

1.  In figure 1 it states "RNAm digestion" instead of "mRNA digestion". Also it reads "Traslation repression" instead of "Translational repression". Please correct.

2. Page 4, Line 140 - "...c-Myc i positively..." what is meant by the i?

3. Page 4, Line 146 - Change "microRNA-7" into "miR-7" to be consistent.

4. Page 6 - Note that Pax-6 refers to the murine protein. I do not believe that is correct, so if one means the human protein use PAX6. The common rule is: 'all capitals' indicate human genes, when referring to nucleic acids (DNA, RNA) one uses italics (i.e. PAX6) when referring to protein one uses straight capitals (i.e. PAX6). Please check and correct when needed. Also check Raf-1 in this context. GeneCards (The Human Gene Database) may be helpful in finding the correct gene designations.

5. Page 17, line 538 - 539 - This Table 1 heading is not in the correct place and needs to be removed.

6. Page 19, line 572 - "Figure analysis was publish previously" should be " Figure analysis was published previously". Please correct.
